# A Case of Early-Stage Gallbladder Cancer, Positive for ALDH1A1, Which Arose from Adenomyomatosis of the Gallbladder

**DOI:** 10.3390/diagnostics12112721

**Published:** 2022-11-07

**Authors:** Yuhei Iwasa, Keisuke Iwata, Mitsuru Okuno, Akihiko Sugiyama, Yoichi Nishigaki, Yosuke Ohashi, Takuji Tanaka, Takuji Iwashita, Masahito Shimizu, Eiichi Tomita

**Affiliations:** 1Department of Gastroenterology, Gifu Municipal Hospital, 7-1 Kashima-cho, Gifu 500-8513, Japan; 2Department of Diagnostic Pathology, Gifu Municipal Hospital, 7-1 Kashima-cho, Gifu 500-8513, Japan; 3Department of Gastroenterology, Gifu University Graduate School of Medicine, 1-1 Yanagido, Gifu 501-1194, Japan

**Keywords:** adenomyomatosis, Rokitansky–Aschoff sinus, BilIN3, stem cell marker, ALDH1A1

## Abstract

Adenomyomatosis (ADM) of the gallbladder is a condition characterized by the proliferation of Rokitansky–Aschoff sinus (RAS), in which the epithelium of the gallbladder extends into the muscular layer, causing a thickening of the gallbladder wall. Although ADM is generally considered not to be a precancerous lesion of gallbladder cancer, there are some reports of cases of gallbladder cancer from ADM. Therefore, the relationship between ADM and gallbladder cancer remains controversial. We herein report a case of early-stage gallbladder cancer, BilIN3 (high grade), arising from ADM that was positive for ALDH1A1, an important marker of stem cells and cancer stem cells.

## 1. Background

Adenomyomatosis (ADM) of the gallbladder is a condition characterized by the proliferation of Rokitansky–Aschoff sinus (RAS), in which the epithelium of the gallbladder extends into the muscular layer or its outer layer, causing a thickening of the gallbladder wall. It was first reported by Jutras [1] in 1960. However, the frequency of gallbladder cancer concomitants with ADM is reported to be 1.4–6.6% [2]. Whether or not ADM itself is a precancerous lesion is controversial. 

Understanding the pathogenesis and carcinogenesis of gallbladder cancer is important. The fifth edition of the World Health Organization’s tumor classification of the digestive system indicates three types of preinvasive neoplasms of the gallbladder [3], including biliary intraepithelial neoplasia (BilIN), pyloric gland adenoma (PGA), and intracholecystic papillary neoplasm (ICPN) [4]. BilINs, which are classified into BilIN-1, BilIN-2, and BilIn-3, are microscopically recognizable as flat or micropapillary preinvasive neoplasms [3]. The increasing grades of biliary intraepithelial neoplasia (BilIN) reflect the multistep carcinogenesis of gallbladder cancer/cholangiocarcinoma, with BilIN-3 representing the stage of carcinoma in situ [5]. Despite the well-defined morphological features in BilIN, the molecular alterations seen during early tumor progression in the biliary tract are poorly understood [6].

The human aldehyde dehydrogenase (ALDH) superfamily comprising 19 isozymes poses important physiological and toxicological functions [7]. The ALDH1A subfamily is also known to play an important role in embryogenesis and development by mediating retinoic acid signaling [8]. The ALDH1A1 gene codes a cytoplasmic enzyme and exerts vital physiological and pathophysiological functions, playing important roles in various chronic diseases, including cancer [9]. ALDH1A1 can induce cancer development via the maintenance of cancer stem cell properties, the modification of the metabolism, and the promotion of DNA repair. 

The expression of ALDH1A1, which acts as a tumor suppressor in certain cancers, is regulated by several epigenetic processes. Since the detoxification of ALDH1A1 often causes the failure of chemotherapy [9], ALDH1A1-targeted therapy has recently been widely used as a cancer chemotherapy [10], although the mechanism by which ALDH1A1 regulates carcinogenesis is not fully understood.

We herein report a surgically treated case of early-stage gallbladder cancer arising from ADM that was positive for ALDH1A1.

## 2. Case Presentation


History of present illness: a 78-year-old woman was referred to our department by her family doctor for further examination of a gallbladder polyp, with gallbladder wall thickening without any symptoms;Medical history: she had type 2 diabetes, hypertension, and hyperuricemia as comorbidities, each of which was treated with medication;Social history: she had a history of smoking 10–15 cigarettes per day and drinking 60 g of alcohol per week;Clinical chemistry: there were no abnormal findings in her clinical chemistry, including with regard to tumor markers;Abdominal ultrasonography: a 7.4-mm stalked polyp on the gallbladder body and wall thickening of the fundus with aggregated multiple cystic structures was seen, suggesting RAS (Figure 1).Contrast-enhanced CT: there were multiple cystic structures at the fundus of the gallbladder and no hyperenhancement of the gallbladder wall or nodules (Figure 2);Contrast-enhanced MRI: there was a 4-mm gallbladder body polyp and no abnormal signals in the thickened wall at the fundus of the gallbladder (Figure 3);
○Based on these imaging findings, we diagnosed the patient with a gallbladder polyp and focal-type ADM; ○The patient chose surgical treatment, and laparoscopic cholecystectomy was performed. The gallbladder was easily dissected from the liver and was able to be removed laparoscopically as usual;Pathological findings: macroscopically, the gallbladder showed a thickened wall containing cystic lesions, suggesting ADM (Figure 4a).


There were no macroscopic neoplastic lesions. Microscopically, there were multifocal cystic lesions with a cluster of large cysts contiguous to the RAS in the fundus of the thickened gallbladder wall (Figure 4b). Among the cystic lesions of varying sizes, the cysts lined with hyperplastic columnar cells (Figure 4c), and a small micropapillary epithelial lesion suggestive of BilIN-1 (low-grade) that was characterized by mild atypia, were noted (Figure 4d). In addition, three different-sized BilIN-3 (high-grade) lesions were present (Figure 4b-cricle/box,e,f) measuring 1 mm (one lesion) and 7 mm (two lesions) in diameter. The lesions showed micropapillary structures lined by atypical cells with hyperchromatic/irregular nuclei, with an increased N/C ratio, complex nuclear stratification, and loss of nuclear polarity (Figure 4b,f). Most of the cysts were covered with poorly atypical columnar-cubical epithelium similar to that of the bile duct or stomach. Immunohistochemically, the cytoplasm of the BillIN-3 lesions was strongly positive for ALDH1A1 (Figure 5a), but negative for S-100P (Figure 6a), SALL4 (Figure 6b), Oct 3/4 (Figure 6c), and glypican 3 (Figure 6d). Interestingly we observed the expression of ALDH1A1 in a few cells which lined in the RAS (Figure 5b), while the gallbladder normal epithelial cells did not express ALDH1A1 (Figure 5c). Nuclear-positive reaction against p53 was also observed in the BilIN-3 lesions (Figure 5d). The MIB-1-positive rate of the BilIN-3 lesions was 60%–70% (hot spot) (Figure 5e).

## 3. Discussion

ADM is a relatively common disease, reportedly found in 2.1–14% of cholecystectomy cases and 0.12–0.49% of medical checkups [11]. The association between ADM and gallbladder cancer is still inconclusive. Ootani et al. [12] reported that 12 cases (4.3%) of gallbladder cancer were found in 279 cases of ADM, all of which were a segmental type, suggesting an association between ADM and gallbladder cancer. Similarly, Nabatame et al. [13] reported a higher coexistence rate of gall bladder cancer in the segmental type of ADM than in other types, speculating that, in segmental ADM, the increase in the intra-gallbladder pressure and bile stasis may cause metaplastic changes in the gallbladder mucosa, resulting in gallbladder cancer development. In both of the aforementioned reports, ADM itself was considered to be an indirect precursor lesion for gallbladder cancer, not a direct cause of cancer.

Mori et al. [14] reported that the relationship between ADM and gallbladder cancer is not clear in their study of 11 cases of the segmental type of ADM complicated with gallbladder cancer, as there were no marked differences in the MIB-1 and p53 expression between the fundal and cervical sites of the segment, and more metaplastic changes were observed in the cervical site, where a low incidence of gallbladder cancer was reported. In contrast, there are several reports of gallbladder cancer arising from within RAS, as seen in these cases [15,16,17,18,19]. In particular, Anno et al. [15] reported that immunohistochemical staining showed that MIB-1 and MUC1 were expressed in the RAS but not in the gallbladder luminal epithelium, suggesting that the RAS itself may be a precursor lesion. In our case, the atypical epithelium was confined within the RAS, and there was no atypical epithelium in the gallbladder lumen or stroma. In addition, the ADM was localized mainly at the fundal side of the gallbladder, suggesting that the RAS itself was the origin of the cancer rather than the result of secondary carcinogenesis triggered by increased gallbladder pressure or bile stasis due to ADM.

In our patient, almost all atypical cells in the BilIN-3 lesions were immunohistochemically positive for ALDH1A1 (Figure 5a). In addition, a few cells in the RAS showed the positive reactivity of ALDH1A1, as shown in Figure 5b. These findings are of interest, as ALDH1A1 is known to play a vital role as a marker of stem cells and cancer stem cells [20]. This is the first report demonstrating that the cytoplasm of the cells in BilIN-3 lesions, as well as cells in RAS, is immunohistochemically positive for ALDH1A1. *ALDH1A1* located in 9q21.13 encodes a homotetramer that is ubiquitously distributed in a variety of adult tissues [20]. ALDH1A1, an important member of the 19-member ALDH family, is able to metabolize reactive aldehydes to their corresponding carboxylic acid derivatives [7,20]. It further plays important physiological and toxicological roles in many areas, including cancer, embryogenesis, and development, by mediating retinoic acid signaling. The overexpression of ALDH1A1 is known to correlate with a poor prognosis and tumor aggressiveness. It is also associated with drug resistance in traditional chemotherapy for cancer [21]. Gallbladder cancer reportedly expresses ALDH1A1 to [22,23], suggesting that ALDH1A1 may be closely related to carcinogenesis in the gallbladder. We expect that the expression of ALDH1, p53, and MIB-1 will increase in number and intensity when BilIN is progressed.

In our case, the diagnosis of gallbladder cancer could not be made preoperatively, as ADM is mainly characterized by the thickening of the gallbladder wall in the imaging findings, and the focal type can be especially difficult to distinguish from gallbladder cancer [24]. While the heterogeneous wall contrast effect on contrast-enhanced CT [25]; wall discontinuity or irregular thickening of the innermost layer, irregular thickening of the outermost layer, loss of a multilayer pattern on the US [26]; and increased signal intensity on diffusion-weighted imaging-MRI are characteristic of gallbladder cancer [27], these findings may not be present in mucosal cancer confined to the RAS, as in our case. In fact, Nakayama et al. [18] reported in 21 literature reports that, among 22 cases of resected RAS-derived gallbladder cancer, only 5 cases of gallbladder cancer could be diagnosed preoperatively. Furthermore, only 2 out of 10 cases of advanced gallbladder cancer were able to be diagnosed [18]. As stated by Cavallaro et al. [28], laparoscopic cholecystectomy does not influence survival if implemented properly. Reoperation should be considered for two aims: R0 resection and clearance of the lymph nodes.

Although there are a few reports of gallbladder cancer arising from RAS [15,16,17,18,19], it is difficult to determine whether or not cancer arises from a RAS. However, our case indicated that adenocarcinoma could develop in the RAS, suggesting the existence of multistep carcinogenesis in the gallbladder through changes in ALDH1A1 expression, which is a cancer stem cell marker [22,23]. Although the question of whether ADM itself is a true precancerous lesion remains controversial, we consider that it may be a putative precursor lesion. In order to confirm this, regular follow-up studies are necessary, with the possibility of malignant transformation being kept in mind. Finally, our case may provide new insights into gallbladder carcinogenesis and associated therapeutic options [7,21].

## 4. Conclusions

We analyzed a case of gallbladder cancer within the RAS that included positive lesions for ALDH1A1.

## Figures and Tables

**Figure 1 diagnostics-12-02721-f001:**
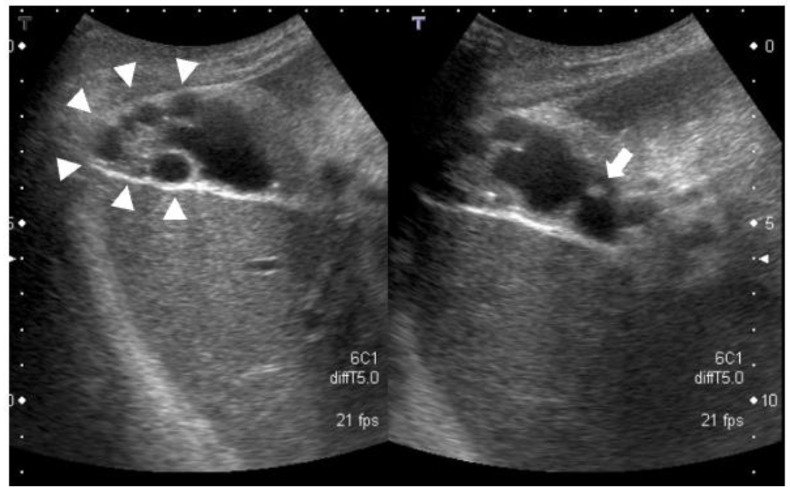
Multiple cystic structures suggest RAS at the fundus of the gallbladder (arrow head). A 7.4-mm stalked polyp on the gallbladder body (arrow).

**Figure 2 diagnostics-12-02721-f002:**
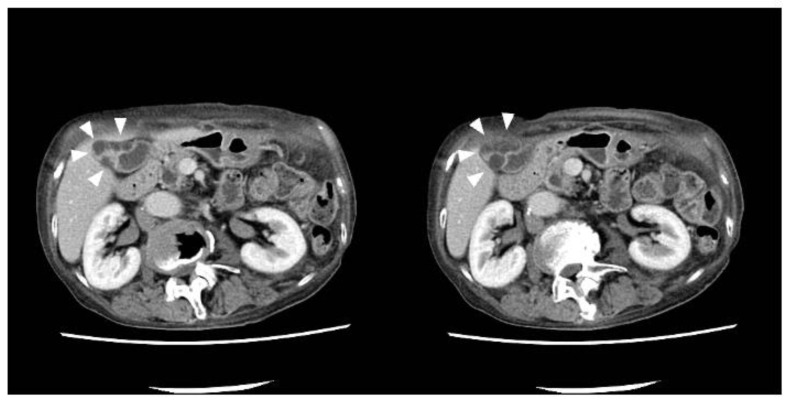
The axial images of contrast-enhanced CT (CE-CT). Multiple cystic structures were described at the fundus of the gallbladder (arrowhead). The contrasting attitude of the fundus of the gallbladder wall was normal, as was the gallbladder neck and body, and there was no irregularity of the gallbladder wall.

**Figure 3 diagnostics-12-02721-f003:**
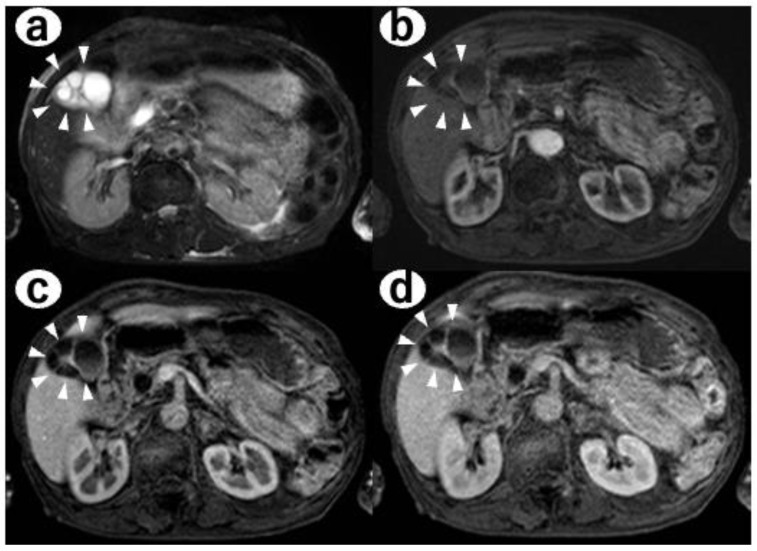
(**a**) T2-weighted image, (**b**) 42 s phases of contrast-enhanced MRI (CE-MRI), (**c**) 72 s phases of CE-MRI, and (**d**) 125 s phases of CE-MRI. There was no abnormal signal in the wall at the fundus of the gallbladder (arrowhead).

**Figure 4 diagnostics-12-02721-f004:**
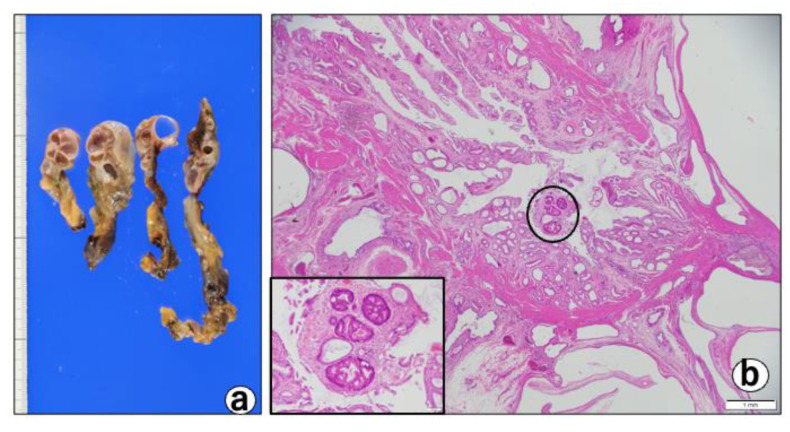
(**a**) Macroscopic view of the cut surfaces of the gallbladder from a specimen removed by laparoscopic cholecystectomy. Note the thickened wall with multilobular cysts. (**b**) Histopathology of the thickened wall indicates adenomyomatosis, with expanded Rokitansky–Aschoff sinuses. H & E stain, Bar = 1 mm. Black circle and box: small foci of BilIN-3, H & E stain, Bar = 100 m. (**c**) A cystic lesion lined by hyperplastic columnar cells. H & E stain, Bar = 50 m. (**d**) A BilIN-1 lesion. H & E stain, Bar = 100 m. (**e**) A large BilIN-3 lesion. H & E stain, Bar = 1 mm. (**f**) High-power view of the large BilIN-3. Note: BilIN-3 consists of atypical cells with enlarged nucleoli. H & E stain, Bar = 50 m. (**g**) Another large BilIN-3 lesion. H & E stain, Bar=1 mm. (**h**) High-power view of the large BilIN-3. Note: BilIN-3 consists of atypical cells with enlarged nucleoli, complex nuclear stratification, and loss of nuclear polarity. H & E stain, Bar = 20 m.

**Figure 5 diagnostics-12-02721-f005:**
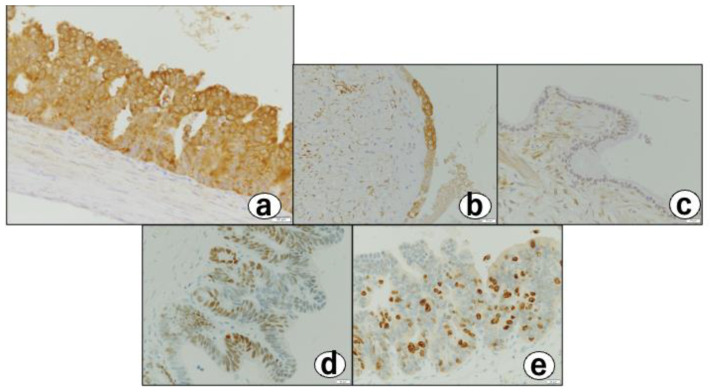
(**a**) Atypical cells in BilIN3 show positive cytoplasm staining of ALDH1A1. (**b**) A few cells lined in a RAS are positive for ALDH1A1. (**c**) Normal appearing gallbladder epithelial cells are negative for ALDH1A1. (**a**–**c**) ALDH1A1-immunohistochemistry using polyclonal rabbit anti-ALDH1A1 antibody (1:100 dilution, ab227948; Abcam, Cambridge, CB2 0AX, UK), Bars = 20 m. (**d**) Nuclear-positive reaction against p53 antibody. Note: Irregular positivity is found in nuclei of atypical cells composing the BilIN-3 lesion. p53 immunohistochemistry with mouse monoclonal anti-human p53 antibody (1:100 dilution, M7001; Dako, Inc., Kyoto, Japan), Bar = 20 m. (**e**) Approximately 65% positive rate of MIB-1 in BilIN-3 lesion. MIB-1 immunohistochemistry with mouse monoclonal anti-human Ki-67 antibody (1:100 dilution, M7240; Dako, Inc., Glostrup, Denmark), Bar = 20 m.

**Figure 6 diagnostics-12-02721-f006:**
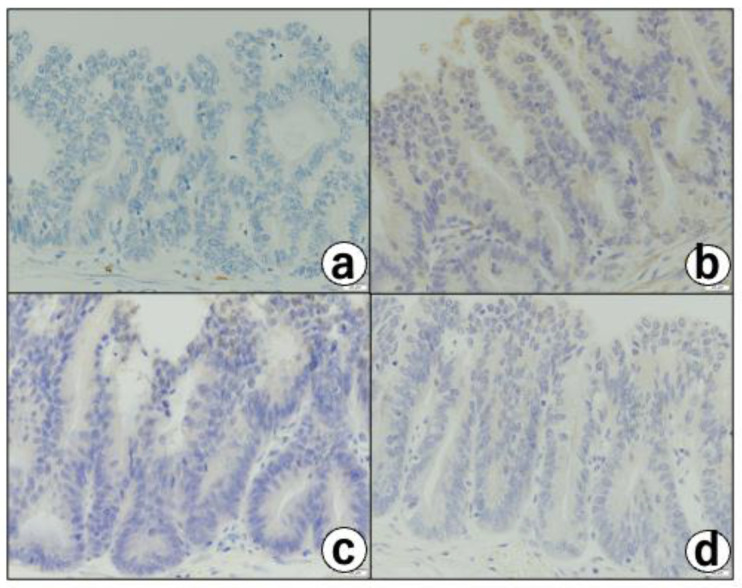
Negative reaction against (**a**) S-100P, (**b**) SALL4, (**c**) Oct 3/4, and (**d**) glypican 3 in atypical cells of BilIN3 using polyclonal rabbit anti-S-100P antibody (1:3000 dilution, HPA019502; Atlas Antibodies, Bromma, Sweden), polyclonal rabbit anti-SALL4 antibody (1:100 dilution, HPA015291, Atlas Antibodies, polyclonal goat anti-Oct 3/4 antibody (1:500 dilution, sc-5279, Santa Cruz Biotechnology In., Santa Cruz, CA, USA), and polyclonal anti-glypican 3 antibody (1:500 dilution, 25175-1-AP, Proteintech Group, Inc., Rosemont, IL, USA), respectively, Bars = 20 μm.

## Data Availability

Not applicable.

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
