# Peer review of "A Case of Early-Stage Gallbladder Cancer, Positive for ALDH1A1, Which Arose from Adenomyomatosis of the Gallbladder"

_diagnostics, 2022, doi:10.3390/diagnostics12112721_

Round 1

Reviewer 1 Report

Iwasa et al report a case of early-stage gallbladder cancer arising from Adenomyomatosis (ADM) that was positive for an important cancer stem cell marker ALDH1A1. This case provides new insight into gallbladder carcinogenesis through changes in the ALDH1A1 expression in the Rokitansky-Aschoff sinus (RAS). The following issues exist with the manuscript as proposed. 

1.    In lines 95-97, the authors mentioned “Macroscopically, the gallbladder showed a normal appearance without a thickened wall containing cystic lesions”. But in Fig.4, it showed the cut-surfaces of the gallbladder with a thickened wall with multi-lobular cysts. The author should further explain it.

2.    It’s better to combine Figures 4-7 into one figure as panels because they are all pathological findings of RAS, ADM, and BilIN. And a border should be used on the Insert for separation (Fig.4b).

3.    It’s better to combine Figures 8-9 into one figure as panels because they are all immunohistochemical results of the BilIN lesions.

4.    In Figures 8-9, the authors only showed the expression of ALDH1A1, p53, and MIB-1 in the BilIN lesions. How about the expression of ALDH1A1, p53, and MIB-1 in the adjacent normal tissue and that in the RAS? The authors should further discuss if the expression of ALDH1A1, p53, and MIB-1 will change after BilIN is developed.

5.    The authors should also show the immunohistochemical results for S-100, SALL4, Oct3/4, and glypican 3, not just mentioned: “data not shown”.

Author Response

We thank the reviewer for the suggestions to improve our manuscript. We agree with the suggestions and have made the changes accordingly.

  1. In lines 95-97, the authors mentioned “Macroscopically, the gallbladder showed a normal appearance without a thickened wall containing cystic lesions”. But in Fig.4, it showed the cut-surfaces of the gallbladder with a thickened wall with multi-lobular cysts. The author should further explain it.

Response 1: Thank you for your comments. Our mistake regarding macroscopic appearance has been corrected.

  1. It’s better to combine Figures 4-7 into one figure as panels because they are all pathological findings of RAS, ADM, and BilIN. And a border should be used on the Insert for separation (Fig.4b).

Response 2: Thank you for your suggestion. We have combined Figures 4-7 into one figure (Figure 4).

  1. It’s better to combine Figures 8-9 into one figure as panels because they are all immunohistochemical results of the BilIN lesions.

Response 3: As you recommended, we have combined Figures 8-9 into one figure (Figure 5).

  1. In Figures 8-9, the authors only showed the expression of ALDH1A1, p53, and MIB-1 in the BilIN lesions. How about the expression of ALDH1A1, p53, and MIB-1 in the adjacent normal tissue and that in the RAS? The authors should further discuss if the expression of ALDH1A1, p53, and MIB-1 will change after BilIN is developed.

Response 4: We did not observe the expression of ALDH1A1 and p53 MIB-1 in the adjacent normal tissue and that in the RAS. Only a few cells in the RAS showed nuclear positivity of MIB-1. This has been added in the text.

  1. The authors should also show the immunohistochemical results for S-100, SALL4, Oct3/4, and glypican 3, not just mentioned: “data not shown”.

Response 5: Thank you for your comments. We have added a figure (Figure 6) showing negative stainabilities of S-100, SALL4, Oct3/4, and glypican 3 in BilIN3 lesion.

Reviewer 2 Report

Well written case report with an adequate discussion

I would like the Authors to provide some “general” details concerning GBC management after pathological diagnosis following cholecystectomy (10.1007/s11605-017-3655-z; 10.1002/bjs.11035)

Congratulations

Author Response

I would like the Authors to provide some “general” details concerning GBC management after pathological diagnosis following cholecystectomy

Response: Thank you for your suggestion. We have added a sentence regarding management of patients with incidental GBC in the text.

Reviewer 3 Report

Dear Author

There are ethical considerations that should be added.

The case report is suggested to be presented based on consort guidelines and written or verbal informed consent should have. 

Author Response

There are ethical considerations that should be added.

The case report is suggested to be presented based on consort guidelines and written or verbal informed consent should have.

Response: Many thanks for your important comments. We have added sentences regarding ethical considerations and verbal informed consent in the text.

Round 2

Reviewer 1 Report

The authors have been responsive to reviewer comments and have supplied an improved manuscript. 

Remaining point:

1. The authors mentioned that “We herein report a case of early-stage gallbladder cancer, BilIN3 (high grade), arising from ADM that was positive for ALDH1A1” in lines 19-20. ADM of the gallbladder is a condition characterized by the proliferation of RAS. That’s the reason I mentioned the authors should further discuss if the expression of ALDH1A1 will change after BilIN is developed (Q4 in review). In the authors’ response 4, the authors mentioned “We did not observe the expression of ALDH1A1……in the RAS” (Also seen in lines 129-131). The authors should further confirm and discuss it.

2. The authors should further show the immunohistochemical results for ALDH1A1 in the RAS and in the adjacent normal tissue to support the authors’ conclusion clearly.

3. In figure 6, the images are too dark to show a clear immunohistochemical staining result.

Author Response

  1. The authors mentioned that “We herein report a case of early-stage gallbladder cancer, BilIN3 (high grade), arising from ADM that was positive for ALDH1A1” in lines 19-20. ADM of the gallbladder is a condition characterized by the proliferation of RAS. That’s the reason I mentioned the authors should further discuss if the expression of ALDH1A1 will change after BilIN is developed (Q4 in review). In the authors’ response 4, the authors mentioned “We did not observe the expression of ALDH1A1……in the RAS” (Also seen in lines 129-131). The authors should further confirm and discuss it.

Reply: Many thanks again for the comments. We have misunderstood your comments. Therefore, we have re-examined all pathological specimens we obtained. We found a few epithelial cells in the RAS are positive for ALDH1A1. This has been added in the text and Figure 5. The findings may support our hypothesis described. Thank you for your suggestion.

  1. The authors should further show the immunohistochemical results for ALDH1A1 in the RAS and in the adjacent normal tissue to support the authors’ conclusion clearly.

As described above, we have added immunohistochemistry of ALDH1A1 of RAS and normal epithelium in Figure 5.

  1. In figure 6, the images are too dark to show a clear immunohistochemical staining result.

Yes, we agree with you. We have replaced Figure 6 by photos of high magnification (bars=20 mm). I think they are clearer than previous photos.

Round 3

Reviewer 1 Report

1. The authors just directly updated the description of the expression of ALDH3A1 in the RAS and that in the gallbladder normal epithelial cells in the manuscript context, without any updated data to support it. Although the authors mentioned Figure 5 has been updated to support it, it's still definitely the same as the old version of Figure 5 now. The expression change of ALDH3A1 between BilIN and RAS is a major conclusion of this manuscript, the authors should show clear and sufficient data to verify it.

2. The authors mentioned they have replaced Figure 6 with photos of high magnification (bar = 22 mm), but it seems like figure 6 is still the old version without any update.

Author Response

  1. The authors just directly updated the description of the expression of ALDH3A1 in the RAS and that in the gallbladder normal epithelial cells in the manuscript context, without any updated data to support it. Although the authors mentioned Figure 5 has been updated to support it, it's still definitely the same as the old version of Figure 5 now. The expression changes of ALDH3A1 between BilIN and RAS is a major conclusion of this manuscript, the authors should show clear and sufficient data to verify it.

Reply: Thank you for the comments. We have added Figure 5b and 5c, which show ALDH1A1 in the RAS and normal gallbladder epithelium, respectively. When you look at Figure 5a, 5 b, and 5c, almost all atypical cells in BilIN3 strongly expressed ALDH1A1 (Figure 5a). However, a few epithelial cells lined in the RAS are positive for ALDH1A1 (Figure 5b). No immunohistochemical expression of ALDH1A1 was observed in normal epithelial cells of the gallbladder (Figure 5c). We consider these findings strongly support our major conclusion.

  1. The authors mentioned they have replaced Figure 6 with photos of high magnification (bar = 22 mm), but it seems like figure 6 is still the old version without any update.

Reply: Thank you for the comments. Figure 6 has been replaced by photos with high magnification (bars=20 micrometer).